# Tissue-Specific Transcriptome and Metabolome Analysis Reveals the Response Mechanism of *Brassica napus* to Waterlogging Stress

**DOI:** 10.3390/ijms24076015

**Published:** 2023-03-23

**Authors:** Bo Hong, Bingqian Zhou, Zechuan Peng, Mingyao Yao, Junjie Wu, Xuepeng Wu, Chunyun Guan, Mei Guan

**Affiliations:** 1College of Agriculture, Hunan Agricultural University, Changsha 410128, China; 2Hunan Branch of National Oilseed Crops Improvement Center, Changsha 410128, China; 3Southern Regional Collaborative Innovation Center for Grain and Oil Crops in China, Changsha 410128, China

**Keywords:** *Brassica napus*, waterlogging stress, metabolome, flavonoid biosynthesis, vitamin B6 metabolism

## Abstract

During the growth period of rapeseed, if there is continuous rainfall, it will easily lead to waterlogging stress, which will seriously affect the growth of rapeseed. Currently, the mechanisms of rapeseed resistance to waterlogging stress are largely unknown. In this study, the rapeseed (*Brassica napus*) inbred lines G230 and G218 were identified as waterlogging-tolerant rapeseed and waterlogging-sensitive rapeseed, respectively, through a potted waterlogging stress simulation and field waterlogging stress experiments. After six days of waterlogging stress at the seedling stage, the degree of leaf aging and root damage of the waterlogging-tolerant rapeseed G230 were lower than those of the waterlogging-sensitive rapeseed G218. A physiological analysis showed that waterlogging stress significantly increased the contents of malondialdehyde, soluble sugar, and hydrogen peroxide in rape leaves and roots. The transcriptomic and metabolomic analysis showed that the differential genes and the differential metabolites of waterlogging-tolerant rapeseed G230 were mainly enriched in the metabolic pathways, biosynthesis of secondary metabolites, flavonoid biosynthesis, and vitamin B6 metabolism. Compared to G218, the expression levels of some genes associated with flavonoid biosynthesis and vitamin B metabolism were higher in G230, such as *CHI*, *DRF*, *LDOX*, *PDX1.1*, and *PDX2*. Furthermore, some metabolites involved in flavonoid biosynthesis and vitamin B6 metabolism, such as naringenin and epiafzelechin, were significantly up-regulated in leaves of G230, while pyridoxine phosphate was only significantly down-regulated in roots and leaves of G218. Furthermore, foliar spraying of vitamin B6 can effectively improve the tolerance to waterlogging of G218 in the short term. These results indicate that flavonoid biosynthesis and vitamin B6 metabolism pathways play a key role in the waterlogging tolerance and hypoxia stress resistance of *Brassica napus* and provide new insights for improving the waterlogging tolerance and cultivating waterlogging-tolerant rapeseed varieties.

## 1. Introduction

In the growth and development of plants, water is a crucial environmental factor [1]. When soil moisture exceeds the maximum water holding capacity in the field, plants will suffer from waterlogging stress, which seriously affects their normal growth. Since the speed of oxygen transmission in water is only 1/10^4^ of that in the air, waterlogging stress is often accompanied by hypoxia or anaerobic stress [2]. Rapeseed (*Brassica napus*) is an important economic crop in the world and one of China’s most important oil crops [3,4]. Oilseed rape lacks aerenchyma, so it is particularly sensitive to waterlogging stress compared to stresses such as light, salinity, and temperature [5]. Rape planting in China is mainly concentrated in the Yangtze River Basin, where the cultivation mode is rice–oilseed rape rotation. Before rape planting, rice is cultivated, which leads to a high groundwater level in the field. Thus, it is easy to accumulate rainwater, which will cause waterlogging stress to rapeseed during the seedling stage [6]. In order to reduce the damage caused by waterlogging stress, agronomic measures such as ditching and drainage, cultivating, turning soil, and increasing the application of organic fertilizers can be adopted in production, but the labor cost is increased [6,7]. Therefore, breeding rapeseed varieties with strong waterlogging tolerance is still the most effective way and one of the most important breeding goals to simplify cultivation and reduce costs. Currently, most research on the waterlogging resistance of rapeseed focuses on screening waterlogging resistant materials and resistance physiology; there is still a large gap in the research on the molecular mechanism.

When plants suffer from waterlogging stress at the seedling stage, it will affect the gas exchange of roots, reduce root hairs, reduce the ability of the root to absorb water and fertilizer, and at the same time, the growth of leaves is inhibited, and the leaf margins turn yellow and dry [8]. On the other hand, deep rice can adapt to the hypoxic environment caused by waterlogging stress by forming adventitious roots and elongating stems [9]. Similarly, rapeseed will also form many adventitious roots to maintain a high energy reserve, thereby alleviating the effects of waterlogging stress injury [5,10]. In addition, waterlogging stress can also seriously affect plant physiology. For example, soluble sugars and malondialdehyde (MDA), among others, all undergo significant changes [11,12].

In addition, previous studies on the waterlogging tolerance of rape roots have been carried out at the transcriptome [13] and proteome [14] levels. Some genes related to secondary metabolism, redox pathways, and transcriptional regulation were significantly enhanced under waterlogging stress. The intracellular life activities are jointly undertaken by many genes, proteins, and metabolites, among which the role of metabolites cannot be ignored. Metabolomics can quantitatively describe metabolite changes in organisms, amplify small changes in gene and protein expression at the functional level, and make detection easier. Metabolomics has been applied to the study of various plants, such as salt tolerance in rape [15], disease resistance in wheat [16], drought tolerance in whitethorn [17], and fruit development of sweet cherry [18]. However, metabolomic studies in rapeseed waterlogging stress are scarce.

In order to screen more waterlogging-tolerant rapeseed varieties and elucidate their potential resistance mechanisms, some rapeseed varieties and inbred lines were selected for pot and field cultivation, and the phenotypes of tolerant and susceptible rapeseed were observed. In addition, after waterlogging stress, the physiological, transcriptomic, and metabolomic changes of leaves and roots of the waterlogging-tolerant *Brassica napus* inbred line G230 and the waterlogging-intolerant *Brassica napus* G218 were analyzed, and the transcriptome and metabolome association analysis were performed to identify key candidate genes, metabolites, and associated pathways.

## 2. Results

### 2.1. Evaluation of Waterlogging Tolerance of Brassica napus

The dead seedling rate of each tested variety or inbred line was counted (Appendix A), and it was found that the dead seedling rate of inbred line G218 was 100%, while the dead seedling rate of inbred line G230 was 0%. In addition, there are also inbred lines G179 and G211 with low seedling mortality rates of 5% and 1.7%, respectively, and inbred lines G233 and G268 with high seedling mortality rates of 96.7% and 98.3%, respectively. We chose G218 and G230 for further experiments.

### 2.2. Morphological and Physiological Changes of Rapeseed Caused by Waterlogging Stress

The simulated field waterlogging stress experiment was carried out at the five leaves seedling stage, and compared with the control; G218 showed chlorosis of new leaves and yellowing or purpling of older leaves when flooded for six days. The phenotype of G230 was visually similar to the control. After ten days of waterlogging, the older leaves of G230 and G218 both turned yellow or purple. At 14 days of waterlogging, the G218 plants died, while the new leaves of the G230 plants grew normally and the old leaves turned yellow (Figure 1A).

In order to observe the waterlogging tolerance of G218 and G230 in field cultivation, we subjected these two materials to field waterlogging stress (the field water content reached 90%; Appendix A) for six days at the seedling stage (five leaves) and then reoxygenated for seven days. As a result, most of the rape G218 plants showed chlorosis of new leaves and yellowing or purpling of old leaves, while no obvious changes in G230 plants were observed, consistent with the potted results (Figure 1B). We conducted a field sampling analysis and found that, compared with the control, G218 significantly decreased the total root length, total root number, shoot fresh weight, fresh root weight, and total fresh weight. Once again, G230 plants showed no significant changes (Table 1).

In order to further analyze the physiological changes brought about by waterlogging stress, we determined several physiological characteristics of rapeseed roots and leaves after waterlogging stress in the field. MDA content is an indicator of plant membrane lipid peroxidation. As shown in Figure 1C, the MDA content of G230 roots did not change significantly under waterlogging stress, indicating that waterlogging damages the root cell membrane less in this variety. However, the MDA content in the root system of G218 was significantly increased. Notwithstanding, the MDA content in leaves of both rapeseed varieties was significantly increased, especially in G218. Thus, compared with the root system, the waterlogging damage to the leaf cell membrane was greater, especially in G218.

Soluble sugars can regulate plant osmotic potential and improve plant stress resistance. The soluble sugar content was significantly elevated in both the roots and leaves of G230 and G218 after waterlogging stress (Figure 1D). In addition, H_2_O_2_ contents in the leaves and roots of G230 and G218 were significantly increased after waterlogging stress, and G218 was significantly higher than G230 (Figure 1E). POD content also increased significantly in G218 leaves and G230 roots (Figure 1F). The results showed that waterlogging stress seriously affected the physiological metabolism of rapeseed seedlings, and compared with G230, waterlogging stress caused more damage to the roots and leaves of G218.

The above results show a significant difference in waterlogging tolerance between the G218 and G230 varieties. Therefore, we regard G218 as a waterlogging-sensitive variety and G230 as a waterlogging-resistant variety.

### 2.3. Transcriptome Sequencing to Identify Differentially Expressed Genes

In order to study the mechanism of rapeseed tolerance and determine the key genes of waterlogging tolerance, transcriptome sequencing was used to analyze the gene expression in leaves and roots of rapeseed G218 and G230 at the seedling stage (five leaves) after six days of field waterlogging stress. Appendix A summarizes the results of three biological replicates of the RNA-seq analysis of each sample in the treatment. Each cDNA library contained 40.663–57.116 million high-quality reads in this experiment. Compared with the rape reference genome, the alignment rate was 77.96–87.17%, the Q20 and Q30 were about 96% and 90%, respectively, and the GC content was about 47%. Based on the PCA analysis (Appendix A), the treated group (WS) and the control group (CK) were clustered together in different regions, and the roots and leaves were clustered together in different regions. This indicated that there were significant differences in the expression profiles of waterlogged and normally grown rapeseed as well as their roots and leaves. The sample correlation heatmap showed a correlation coefficient greater than 0.8, which indicated good biological reproducibility within the same group (Appendix A). The above results show that the sequencing data are reliable and meet the needs of the subsequent analysis.

By comparing root (R) and leaf (L) samples of the same rapeseed under different conditions (control, CK and waterlogging stress, WS) and different rapeseed varieties (G218 and G230) under the same conditions, we constructed eight comparison groups: G218CKL vs. G230CKL, G218CKR vs. G230CKR, G218CKL vs. G218WSL, G218CKR vs. G218WSR, G230CKL vs. G230WSL, G230CKR vs. G230WSR, G218WSL vs. G230WSL, G218WSR vs. G230WSR. Respectively, in G218CKL vs. G230CKL, G218CKL vs. G218WSL, G230CKL vs. G230WSL, G218WSL vs. G230WSL, G218CKR vs. G230CKR, G218CKR vs. G218WSR, G230CKR s. G230WSR, and G218WSR vs. G230WSR identified 700 (394 down-regulated, 306 up-regulated), 17,591 (8225 down-regulated, 9366 up-regulated), 16,606 (7734 down-regulated, 8872 up-regulated), 1695 (773 down-regulated, 922 up-regulated), 560 (198 down-regulated, 362 up-regulated, 16,226 (7846 down-regulated, 8380 up-regulated), 15,858 (6625 down-regulated, 9233 up-regulated), and 1577 (1372 down-regulated, 205 up-regulated) DEGs (Figure 2A). The DEGs of G218CKL vs. G230CKL were significantly lower than those of G218WSL vs. G230WSL, and the DEGs of G218CKR vs. G230CKR were also significantly lower than those of G218WSR vs. G230WSR, indicating that the number of DEGs in both varieties increased under waterlogging stress. In addition, the number of DEGs in G218CKL vs. G218WSL was significantly higher than that in G230CKL vs. G230WSL, and the DEGs in G218CKR vs. G218WSR were also significantly higher than those in G230CKR vs. G230WSR, indicating that the root and leaf responses of G218 were more severe than those of G230 under waterlogging stress. Therefore, the overall change in G218 was larger than in G230. The heatmap in Figure 2B summarizes the expression of all identified DEGs. We used Venn diagrams to compare up- and down-regulated DEGs in different comparisons, and in the leaf samples, six and one DEGs were up- and down-regulated in all four comparison groups (Figure 2C,D). In the root samples, zero and two DEGs were up- and down-regulated in all four comparison groups, respectively (Figure 2E,F).

### 2.4. Differential Gene GO and KEGG Analysis

We classified the 50 most enriched GO terms among all DEGs in response to waterlogging with different comparisons. As shown in Figure 3 and Appendix A, there were three GO categories: biological process (BP), cellular component (CC), and molecular function (MF). In the leaf samples, G218CKL vs. G218WSL and G230CKL vs. G230WSL were enriched in the three GO categories, and the enrichment entries were the same (Figure 3A,C). Entries enriched in the MF category were “oxidoreductase activity, acting on peroxide as acceptor” and “peroxidase activity”. In the CC category, “photosystem” was the most enriched. In the BP category, “photosynthesis, light reaction” was the most enriched. G218CKL vs. G230CKL was also enriched in all three GO classes (Appendix A), and in the BP class, the “reactive nitrogen species metabolic process” was the most enriched. In the MF category, the most enriched was “inorganic anion transmembrane transporter activity”. In the CC category, however, there were only two functional enrichments: “integral component of plasma membrane” and “central vacuole”. In G218WSL vs. G230WSL, DEGs were only enriched in BP and MF classes (Appendix A). The most enriched in the BP class were “flavonoid metabolic process” and “flavonoid biosynthetic process”. The most enriched in the MF category were “inorganic anion transmembrane transporter activity” and “oxidoreductase activity, acting on paired donors, with incorporation or reduction of molecular oxygen, NAD (P) H as one donor, and incorporation of one atom of oxygen”. In the root samples, we found that the GO enrichment profiles of G230CKR vs. G230WSR and G218CKR vs. G218WSR comparisons were similar to those of G218CKL vs. G218WSL and G230CKL vs. G230WSL (Figure 3B,D). However, G218CKR vs. G230CKR and G218WSR vs. G230WSR were different from the leaf samples. In the G218CKR vs. G230CKR comparison, DEGs were only enriched in BP and MF. The most enriched in the BP category was “defense response, incompatible interaction”, and the most enriched in MF are “inorganic anion transmembrane transporter activity” and “oxidoreductase activity, acting on paired donors, with incorporation or reduction of molecular oxygen, NAD (P) H as one donor, and incorporation of one atom of oxygen”(Appendix A). For G218WSR vs. G230WSR, the most enriched BP category was “cellular response to phosphate starvation”, and the most enriched in MF were “disulfide oxidoreductase activity”, “oxidoreductase activity, acting on a sulfur group of donors,” and “protein disulfide oxidoreductase activity”. There was only one feature enrichment of “cell surface” in the CC category (Appendix A).

According to the KEGG analysis results, the pathways of differential gene enrichment in roots and leaves were also similar. Pathways with more gene mappings are metabolic pathways, biosynthesis of secondary metabolites, starch and sucrose biosynthesis, and phenylpropane biosynthesis. In addition, many genes were mapped to the plant MAPK signaling pathway, tissue synthesis of ubiquinone and other terpenoid quinones, glyoxylate and dicarboxylic acid metabolism, and vitamin B6 metabolism, among others (Appendix A).

### 2.5. Differentially Expressed Metabolites (DEMs) of Two Rape Varieties under Waterlogging Stress

In order to further analyze the metabolites of rape under waterlogging stress, a broad-targeted metabolome was used for the analysis. A total of 963 metabolites were obtained from all samples (Figure 4A,B), and we used a PCA analysis (Appendix A) to investigate the mass spectrometry data of the treatment groups. The results showed that all the biological replicate data points of the roots and leaves of the treatment groups were clustered together, and the two plant materials and the sample data points of different treatments could be clearly distinguished in space and concentration differences.

We detected 224 (24 up-regulated, 200 down-regulated), 559 (290 up-regulated, 269 down-regulated), 586 (3947 up-regulated, 200 down-regulated), and 156 (66 up-regulated, 90 down-regulated) DEMs, respectively, in the leaf sample comparative pairs G218CKL vs. G230CKL, G218CKL vs. G218WSL, G230CKL vs. G230WSL, and G218WSL vs. G230WSL. In the root sample comparisons of G218CKR vs. G230CKR, G218CKR vs. G218WSR, G230CKR vs. G230WSR, and G218WSR vs. G230WSR, 258 (49 up-regulated, 209 down-regulated), 582 (284 up-regulated, 298 down-regulated), 575 (350 up-regulated, 225 down-regulated) and 119 (74 up-regulated, 45 down-regulated) DEMs were detected, respectively. (Figure 4A). We used Venn diagrams to analyze up- and down-regulated DEMs in each of the four comparisons in roots and leaves and found that four common DEMs were significantly up-regulated and 14 common DEMs were significantly down-regulated in the four comparisons in the leaves. In addition, G218CKL vs. G218WSL had 226 and 151 specific DEMs up- and down-regulated compared with G230CKL vs. G230WSL (Figure 4C,D). None of the common DEMs were significantly up-regulated in the four comparisons of roots, but 13 were significantly down-regulated. G218CKR vs. G218WSR had 214 and 131 specific DEMs up- and down-regulated compared with G230CKR vs. G230WSR (Figure 4E,F). The KEGG pathway enrichment analysis showed that DEMs with different comparative combinations were enriched in multiple pathways, such as phenylpropane biosynthesis and flavonoid biosynthesis. The top twenty KEGG enrichment entries are shown in Appendix A. At the same time, we found that the main enrichment entries of up-and down-regulated DEMs in G218CKL vs. G218WSL and G230CKL vs. G230WSL were similar. The up-regulated DEMs were mainly related to phenylpropane biosynthesis, flavonoid biosynthesis, flavonoid and flavonol biosynthesis, ascorbic acid, and aldose metabolism (Figure 5A,C), while the down-regulated DEMs were mainly related to linoleic acid metabolism, starch, and related to sucrose metabolism, biosynthesis of various secondary metabolites—part 2, glycolysis/gluconeogenesis, biosynthesis of unsaturated fatty acids, arachidonic acid metabolism, and alpha-linolenic acid metabolism (Figure 6A,C). DEMs up-regulated in G218CKR vs. G218WSR were significantly enriched in the biosynthesis of secondary metabolites, phenylpropane biosynthesis, flavonoid biosynthesis, and flavonoid and flavonol biosynthesis (Figure 5B). The up-regulated DEMs in G230CKR vs. G230WSR were significantly enriched in phenylpropane biosynthesis, flavonoid biosynthesis, flavonoid and flavonol biosynthesis, ascorbic acid and aldose metabolism, and glutathione metabolism (Figure 5D). Both G218CKR vs. G218WSR and G230CKR vs. G230WSR down-regulated DEMs were significantly enriched in linoleic acid metabolism, purine metabolism, arachidonic acid metabolism, and α-linolenic acid metabolism (Figure 6B,D).

### 2.6. Combined Transcriptome and Metabolome Analysis

In G218 and G230, many DEGs and DEMs were enriched in different KEGG pathways, including metabolic pathways, biosynthesis of secondary metabolites, phenylpropane biosynthesis, flavonoid biosynthesis, and vitamin B6 metabolism. Figure 7 shows the Top 20 KEGG-enriched pathways by *p*-value. In both lines, root and leaf DEGs and DEMs were significantly enriched for many antioxidant-related pathways, including flavonoid biosynthesis and vitamin B6 metabolism. Other enrichment pathways are mostly related to nutrient metabolism, such as glyceride, starch, and sucrose metabolism. The physiological experiments showed that waterlogging greatly impacted the rapeseed’s reactive oxygen species scavenging system, so we analyzed the two antioxidant-related pathways of flavonoid biosynthesis and vitamin B6 metabolism.

### 2.7. Changes in Flavonoid Biosynthesis under Waterlogging Stress

During the biosynthesis of flavonoids, extremely rich flavonoid compounds are produced, and these compounds have significant effects on plant antioxidants. Through the analysis of the transcriptome and metabolome of rapeseed under waterlogging stress, we found that the flavonoid biosynthesis pathway was significantly enriched in the roots and leaves of G218 and G230, and we analyzed the main DEGs and DEMs involved in this pathway. Leaf samples had 39 DEGs, of which 21 were up-regulated, 12 were down-regulated in G230CKL vs. G230WSL; 24 were up-regulated, and 11 were down-regulated in G218CKL vs. G218WSL. In root samples, there were 33 DEGs, of which 21 DEGs were up-, and ten were down-regulated in G230CKR vs. G230WSR; 17 DEGs were up- and 11 were down-regulated in G218CKR vs. G218WSR. These DEGs include *CHS*, *CHI*, *FLS*, *DFRA*, and *LDOX*. For example, the two homologous genes of *CHI* (BnaA01G0107500ZS and BnaC01G0131000ZS) were up-regulated about 3.12-fold and 3.16-fold in the leaves of G230 after waterlogging stress and up-regulated by about 2.90-fold and 3.01-fold in the roots. There were no significant changes in G218 (Figure 8B).

The metabolome analysis showed that compared with the control, the flavonoid biosynthesis pathway of roots and leaves after waterlogging stress involved significant changes in five metabolites such as naringenin chalcone, naringenin, epiafzelechin, eriodictyol, and dihydroquercetin. As shown in Table 2, among these metabolites, four were significantly up-regulated in G230WSL, and none were down-regulated. Three were significantly up-regulated in G218WSL, and none were significantly down-regulated. Among them, epiafzelechin is the most variable in G230WSL (FC is about 11.546), while the most variable in G218WSL is eriodictyol (FC is about 10.766). Notably, epiafzelechin and naringenin changed significantly in G230WSL (FC was 11.546 and 1.445, respectively), but not in G218WSL. In addition, two metabolites were up-regulated, and none were down-regulated in G230WSR, while five were up-regulated and none were down-regulated in G218WSR. Among them, naringenin chalcone, naringenin, and epiafzelechin were significantly up-regulated in G218WSR (FC were 1.506, 1.892, 11.427, respectively), but there was no significant change in G230WSR.

### 2.8. Changes in Vitamin B6 Metabolism in Rapeseed under Waterlogging Stress

In addition to flavonoid biosynthesis, some DEGs and DEMs were enriched in vitamin B6 metabolic pathways. Twenty-two differential genes were found in the two comparisons of G218CKL vs. G218WSL and G230CKL vs. G230WSL. Twenty-one differential genes were found in the two comparisons of G218CKR vs. G218WSR and G230CKR vs. G230WSR. These genes include *PLR1*, *PDX*, and *PNPO*. In G218CKL vs. G218WSL, 14 genes were significantly up-regulated, and three were significantly down-regulated. In G230CKL vs. G230WSL, 16 and five genes were significantly up- and down-regulated. For example, the expression levels of two homologous genes (BnaA04G0245700ZS and BnaC04G0560700ZS) of the *PDX1.1* gene were up-regulated about 10.56-fold and 14.12-fold in G218WSL, and about 10.72-fold and 14.09-fold in G230WSL. In G218CKR vs. G218WSR, 17 and seven genes were significantly up- and down-regulated. Seventeen and eight genes were significantly up- and down-regulated in G230CKR vs. G230WSR. Among the up-regulated genes, two homologous of the *PDX1.1* gene (BnaA04G0245700ZS and BnaC04G0560700ZS) were included. It was up-regulated by 10.89-fold and 12.55-fold in G218WSR and 11.14-fold and 14.83-fold in G230SR. In addition, we also found that two homologous genes of the *PDX2* gene (BnaA02G0098300ZS and BnaA02G0098100ZS) were significantly up-regulated in G230WSL but not significantly changed in G218WSL (Figure 9B).

The metabolome results showed that three metabolites, including pyridoxal, pyridoxal phosphate, and pyridoxine, changed significantly after waterlogging stress (Table 2). Notably, pyridoxal phosphate was significantly down-regulated in G218WSL and G218WSR compared with controls (FC approximately −14.201 and −13.812), while there was no significant change in G230WSL and G230WSR.

### 2.9. qRT-PCR Validation

To validate the differential expression results obtained from the transcriptome analysis, we examined the relative expression levels of four of these differential genes (*PDX2*, *CER26L*, *CHI*, *DFR*) using qRT-PCR. These genes are involved in flavonoid biosynthesis and vitamin B6 metabolism. The expression patterns of DEGs obtained by RNA-seq and qRT-PCR were highly consistent, indicating the reliability of RNA-seq results (Figure 10).

### 2.10. Effect of External Application of Vitamin B6 on Waterlogging Stress in Rapeseed

In order to verify the role of the vitamin B6 metabolic pathway in rapeseed under waterlogging stress, we sprayed rapeseed leaves with a 1 mg/L vitamin B6 solution when rapeseed seedlings grew to five leaves and then continued the waterlogging stress. It was found that on the 10th day of waterlogging stress, on the waterlogging-tolerant rapeseed G230, the phenotype of yellowing of old leaves began to appear, and on the 20th day of waterlogging, the G230 plants were dwarfed, and their biomass decreased. On the other hand, the waterlogging stress-sensitive rape G218 began to show yellowing and dropping phenotypes on the 14th day of the stress, and G218 seedlings died on the 20th day of the stress (Figure 11A). Compared with plants not sprayed with the vitamin B6 solution (Figure 1A), the phenotype of G230 was not significantly changed, but the time of the first yellowing of G218 leaves was delayed by eight days, and the time of plant death was delayed by six days. Then, we analyzed the MDA, POD, and H_2_O_2_ of leaves and roots under waterlogging stress for 6 days after spraying vitamin B6. Compared with the treatment without exogenous VB6, spraying exogenous VB6 before waterlogging stress significantly increased the POD content in G218 and G230 leaves, and significantly decreased the MDA and H_2_O_2_ content in G218 and G230 leaves, especially in G218. The MDA in roots did not change significantly in G218 and G230, but the POD in roots of G230 and H_2_O_2_ in roots of G218 decreased significantly. The reason may be that exogenous VB6 has a better alleviating effect on the waterlogging stress of G218 (Figure 11B). The results showed that foliar spraying of vitamin B6 could effectively improve the waterlogging tolerance of *Brassica napus*.

## 3. Discussion

At present, transcriptomics has been widely used in the study of waterlogging tolerance of plants such as rapeseed, kiwifruit, alfalfa, and wheat, but metabolomics have rarely been used [19,20,21,22]. With the rapid development of modern molecular biology, the research on the mechanism of plant water resistance has reached the level of multi-omics research. As a result, integrating transcriptomic and metabolomic analyses has become a standard tool, providing a powerful and efficient method for identifying waterlogging tolerance genes and mining their related metabolites. However, omics research requires unique germplasm resources and genetic backgrounds [15], especially for *Brassica napus*, a cruciferous allotetraploid plant. In our study, two *Brassica napus* inbred line varieties (G218 and G230) were selected for transcriptomics and metabolomics after screening indoor simulated waterlogging stress and field waterlogging stress. G218 and G230 are both high oleic acid inbred lines. G230 grows well in the rainy and humid climate of Changsha, Hunan Province, China, and is unaffected by waterlogging stress, while G218 is more sensitive to waterlogging stress. Our results showed that G230 and G218 had significant differences in waterlogging tolerance at the seedling stage, so these two inbred lines were ideal genetic resources for studying the waterlogging tolerance mechanism of rapeseed.

Waterlogging stress affects rapeseed germination, emergence, and seedling growth, causing phenotypic changes and physiological and metabolic activities [19] In this study, the phenotypes of rapeseed changed significantly after simulated waterlogging stress and field waterlogging stress. In the pot and field experiments, the leaves of waterlogging-intolerant G218 turned yellow or purple more seriously than those of the waterlogging-tolerant G230. Furthermore, the root length, fresh and dry weight, and root–shoot ratio of G230 plants were higher than those of G218 after field waterlogging stress. Physiological indicators such as the MDA, POD, and H_2_O_2_ were measured to explore the reasons behind the morphological differences between these varieties. When plants are subjected to stress, reactive oxygen species (ROS) can accumulate, causing damage to plant cells [23]. After waterlogging stress, the MDA content in roots and leaves of G218 increased significantly, while the accumulation of the MDA content in G230 only occurred in leaves and in smaller quantities than in G218. The MDA can reflect the degree of membrane damage caused by stress, and our results showed that compared with the waterlogging-tolerant rapeseed G230, the root and leaf membrane damage of the waterlogging-intolerant rapeseed G218 was more serious. Soluble sugars not only play an energy role but are also a class of osmotic regulators, and their accumulation in plants can increase the osmotic potential and maintain membrane stability, thereby improving plant tolerance to stress [24]. The contents of soluble sugar in the roots and leaves of G218 and G230 were significantly increased, and compared with the roots of G218, the accumulation of soluble sugar content in the roots of G230 was more significant. Furthermore, under waterlogging stress, we observed a significant accumulation of reactive oxygen species H_2_O_2_ in rapeseed, and the accumulation of H_2_O_2_ was more pronounced in root and leaf tissues of G218 compared with G230. At the same time, the POD activity was observed differentially in G218 and G230 roots and leaves. The POD activity did not change significantly in G218 roots and G230 leaves but was significantly enhanced in G218 leaves and G230 roots. Therefore, our study showed that G218 roots and leaves were more damaged by waterlogging stress, and G230 could better alleviate and adapt to the damage caused by waterlogging stress by regulating the MDA, osmoregulatory substances, and antioxidant system. Moreover, root and leaf tissues seemed to adapt to waterlogging stress through different response mechanisms. In field cultivation, hypoxia caused by waterlogging stress directly affects the roots of rape [25]. Based on the present results, we speculate that G230’s root tissue ability to regulate the physiological changes under waterlogging stress better makes this variety tolerant.

Previous studies have shown that the waterlogging tolerance of rapeseed is regulated by multiple genes [26,27], and the molecular mechanism of the tolerance can be explored more comprehensively using omics technology. For example, through proteomics and transcriptomics, it was found that rapeseed resistance to waterlogging stress involves various pathways at multiple regulatory levels, such as metabolic processes, signal transduction, and redox processes [13,14]. The above provides important clues for studying the mechanism of rapeseed waterlogging tolerance. However, the response to waterlogging stress in rapeseed is a complex response mechanism involving multiple genes, pathways, and metabolic processes, and the current research in this field is still quite limited. A combined analysis of transcriptomics and metabolomics helps to reveal complex responses to waterlogging stress at the whole plant level. Previous studies have shown that many genes and transcription factors related to redox, secondary metabolism, reactive oxygen species scavenging, and transcriptional regulation play key roles in coping with plant waterlogging stress, such as *GPX*, *GST*, *WRKY*, and *ERF* [9,13,21,28,29].

The analysis of DEGs and DEMs identified many genes or metabolites related to antioxidant reduction under waterlogging stress, providing a deeper understanding of the mechanism of rapeseed tolerance. A KEGG enrichment analysis showed that DEGs and DEMs were mainly involved in metabolic processes, secondary metabolism, phenylpropane biosynthesis, flavonoid biosynthesis, and some pathways related to nutrient metabolism. Furthermore, it is worth noting that the vitamin B6 metabolic pathway was more significantly enriched in root and leaf tissues of G230 compared with G218. Besides the physiological changes of rapeseed under waterlogging stress, we paid special attention to two antioxidant-reduction-related pathways: flavonoid biosynthesis and vitamin B6 metabolism [30,31].

Flavonoids can effectively scavenge reactive oxygen species and free radicals in plants and alleviate the damage caused by oxidative stress [32,33]. Studies have shown that flavonoids positively correlate with plant extracts’ antioxidant capacity [34,35]. For example, flavonoids in sunflowers (JHK) are effective against the ROS inhibitory effect [36]. The present study found that genes and metabolites related to the flavonoid biosynthetic pathway were differentially expressed in different rapeseeds (G230 and G218) and tissues (root and leaf). These genes include *CHI*, *FLS*, *DFR*, and *LDOX*, among others. *CHI* is a key enzyme in the early stage of flavonoid biosynthesis; it catalyzes the isomerization of naringenin chalcone to naringenin, which can be used as a substrate to enter other metabolic pathways for the synthesis of different flavonoids [37]. Under waterlogging stress, the *CHI* transcript levels in the leaves and roots of the waterlogging-tolerant rapeseed G230 were higher than those of the sensitive rapeseed G218. In fact, the two homologous genes of *CHI* (BnaA01G0107500ZS and BnaC01G0131000ZS) were significantly up-regulated in root and leaf tissues in G230 but not in G218. Those may be the main *CHI* genes that respond to waterlogging stress, but further studies are needed to determine their regulatory mechanism. Furthermore, significant down-regulation of *FLS* was detected in G230 and G218 after waterlogging stress, whereas *DFR* and *LDOX* were significantly up-regulated. Flavonol synthase (*FLS*) can catalyze the production of kaempferol and quercetin from two dihydroflavonols, dihydrokaempferol and dihydroquercetin, thereby increasing the accumulation of flavonoids. Dihydroflavonol 4-reductase (*DFR*) can catalyze the synthesis of white anthocyanins from dihydroflavonols. Then, the white anthocyanins can serve as substrates for leucoanthocyanidin dioxygenase (*LDOX*) to catalyze the synthesis of anthocyanins. There are two ways to increase anthocyanins in plants. The first is to increase the accumulation of dihydroflavonols to promote the accumulation of anthocyanins. The second is to reduce the activity of *FLS* and increase the activity of *DFR*, thereby reducing the accumulation of flavonoids and increasing the synthesis of anthocyanins [38,39] (Figure 8A). In our study, a large number of *FLS* genes were significantly down-regulated in the root and leaf tissues of G230 and G218. In contrast, some *DFR* and *LDOX* genes were significantly up-regulated. Therefore, compared with kaempferol and quercetin, rapeseed may preferentially synthesize leucocyanidins and anthocyanidins under waterlogging stress. Although the metabolome results showed no significant changes in leucocyanidins and anthocyanidins in G218 and G230, a proanthocyanin downstream of anthocyanidins and epiafzelechin accumulated significantly in the leaves of G230 and the roots of G218 (Table 2). This may be one of the reasons why G218 and G230 have different responses to waterlogging stress.

Vitamin B6 is a coenzyme of many key enzymes and participates in multiple enzymatic reactions, playing an important role in amino acid, fatty acid, and glucose metabolism [40,41,42]. Studies have shown that vitamin B6 can also scavenge reactive oxygen species and protect plants from photooxidative stress [31]. Moreover, exogenous vitamin B6 protected Arabidopsis protoplasts from singlet oxygen (^1^O_2_)-induced cell death [43]. In this study, after waterlogging stress, vitamin B6 metabolic pathways were more significantly enriched in waterlogging-tolerant rapeseed G230, and some genes and metabolites were significantly changed. The de novo synthesis of vitamin B6 in plants relies on two proteins, *PDX1* and *PDX2*, which function as transglutaminase enzymes and produce pyridoxal phosphate (PLP) from glycolytic intermediates and the pentose phosphate pathway [44,45] (Figure 9A). We found that the *PDX2* gene was significantly up-regulated in G218 root tissue, G230 root tissue, and leaf tissue after waterlogging stress. Two genes of *PDX1.1* were significantly up-regulated in root and leaf tissues in both G218 and G230 but more significantly in G230. In addition, the metabolite PLP was significantly down-regulated in both root and leaf tissues of G218, while there was no significant change in G230. We speculate that these genes and metabolites may be related to the difference in waterlogging tolerance between the two rapeseed varieties. Since exogenous vitamin B6 can protect protoplasts from singlet oxygen stress, can it alleviate the hypoxic stress caused by waterlogging stress? Under waterlogging stress, we found no obvious difference in the phenotype of G230 compared with the control by applying vitamin B6 to rape leaves, but the difference was significant in G218. After the vitamin B6 treatment, G218 had obvious yellow leaves on the 14th day of waterlogging stress, eight days later than the control, and the plants died on the 20th day, six days later than the control. These results suggest that vitamin B6 plays an important role in rapeseed resistance to waterlogging and hypoxia stress.

## 4. Materials and Methods

### 4.1. Screening of Waterlogging Tolerant Brassica napus

An amount of 32 inbred lines and varieties of *Brassica napus* were stored in Hunan Agricultural University. In order to screen rapeseed resistant to waterlogging stress, a pot experiment was conducted in September 2021 to simulate waterlogging stress, and the method was referred to by Li Yun et al. [46]. An amount of 32 seeds of *Brassica napus* were sown in small potted plants with nutritive soil. The inner diameter of the basin bottom was 6 cm, the inner diameter of the mouth is 7.6 cm, and the height is 10 cm. Each basin contains about 100 g of nutrient soil. Before loading soil, small holes were drilled at the bottom of the basin and place them in a watertight container with a length of 100 cm, a width of 65 cm, and a height of 30 cm when simulating waterlogging stress. During drainage treatment, the small pot was taken out. When the seeds emerged normally until the cotyledon fully stretched, 20 seedlings with uniform growth were kept in each pot and treated with waterlogging stress for 6 days (waterlogging until the water surface was higher than the cotyledon). After 6 days of treatment, reoxygenated and normal growth for 7 days was performed to calculate the death rate. The whole experiment was repeated three times.

### 4.2. Plant Material and Waterlogging Stress Treatment

The inbred lines G218 and G230 were used for further experiments. We designed two waterlogging stress experiments: pot-simulated and field. First, in May 2021, G230 and G218 seeds were sown in pots outside at the National Oil Crops Improvement Center in Hunan. The pot specification and simulated waterlogging stress method are the same as Section 4.1. Then, when the rape seedlings reached five leaves, they were submerged in water above the soil surface for 20 days. The experiment was repeated three times.

For the second experiment, seeds were planted in the experimental field of Yunyuan Base of Hunan Agricultural University in October 2021 (28°12′ N, 112°59′ E). Each material is evenly planted in 10 rows with 10 plants in each row, with a row spacing of 30 cm and a plant spacing of 20 cm. Waterlogging stress was applied when the seedlings reached five leaves. Then, the field was flooded with water about 2 cm above the soil surface for six days (the outlet of the field was blocked when waterlog stressed).

Reoxygenation occurred after waterlogging stress (open water outlet to release field water). After seven days of reoxygenation, the fourth leaf and the taproot of rapeseed were collected simultaneously from the field and rinsed with distilled water. All samples were immediately snap-frozen in liquid nitrogen and stored at −80 °C for future use for physiological, transcriptomic, and metabolomic analyses. Three plants in each treatment and the control groups of the two rapeseed varieties were used as replicates for measuring the root length, root tip number, and fresh weight. The control plants were grown in the regular field during the same period. Thus, the whole experiment comprised of eight groups (Table 3).

### 4.3. Determination of Physiological Indicators

To determine the content of soluble sugar, malondialdehyde (MDA), peroxidase (POD), and hydrogen peroxide (H_2_O_2_), 0.1 g of rapeseed fresh leaves and roots were used. These physiological indicators were evaluated according to the instructions through a plant soluble sugar content assay kit (D799392), malondialdehyde content assay kit (D799762), peroxidase activity assay kit (D799592) and hydrogen peroxide content assay kit (D799774) (Shenggong Biotechnology Co., Ltd., Shanghai, China). The root length and the number of root tips were analyzed using a root scanning system, in which all the main roots and lateral roots were intercepted and placed into an EPSON V700 scanner, and analyzed using the WinRhizo system (Regent, Vancouver, BC, Canada). Waterlogging tolerance coefficient (WTC) is the ratio of the measured value of the treatment to the measured value of the control check. This was repeated three times and the average was taken.

### 4.4. RNA Extraction, Sequencing, and Transcriptome Data Analysis

Total RNA samples were isolated using the RNAprep Pure Plant Plus Kit (Tiangen Biotech Co., Ltd., Beijing, China) according to the manufacturer’s instructions. A 1% agarose gel and a NanoPhotometer^®^ spectrophotometer (IMPLEN, Westlake Village, CA, USA) were used to test the integrity of the total RNA. The cDNA library construction and RNA-seq were performed by Wuhan MetWare Biotechnology Co., Ltd. (Wuhan, China). A total of 1 μg RNA per sample was used to generate a cDNA library using the NEBNext^®^ UltraTM RNA Library Prep Kit (NEB, Ipswich, MA, USA), following the manufacturer’s instructions. Briefly, mRNA was purified from total RNA using poly-T oligo attached magnetic beads. Fragmentation was carried out using divalent cations under an elevated temperature in the NEBNext First Strand Synthesis Reaction Buffer (5X). First strand cDNA was synthesized using a random hexamer primer and M-MuLV Reverse Transcriptase (RNase H-). Second strand cDNA synthesis was subsequently performed using DNA Polymerase I and RNase H. Remaining overhangs were converted into blunt ends via exonuclease/polymerase activities. After adenylation of 3′ ends of DNA fragments, the NEBNext Adaptor with a hairpin loop structure was ligated to prepare for hybridization. In order to select cDNA fragments of preferentially 250~300 bp in length, the library fragments were purified with the AMPure XP system (Beckman Coulter, Beverly, MA, USA). The libraries were then sequenced on an Illumina Novaseq platform and paired-end reads were generated. Clean reads were extracted with base-pair qualities in the Q ≤ 20 using custom Perl scripts and mapped to the reference genome (ZS11: https://www.ncbi.nlm.nih.gov/assembly/GCF_000686985.2/, accessed on 19 January 2022) using the default parameters of the HISAT v2.1.0 software. Fragments per kilobase of transcript per million mapped reads (FPKM) were used to quantify gene expression levels. An absolute log2 (fold change) ≥ 2 and false discovery rate (FDR) < 0.05 were used as thresholds for the identification of differentially expressed genes (DEGs) using DESeq2 software [47,48]. A gene Ontology (GO) enrichment analysis was performed using the GOSeq R package (corrected *p*-value < 0.05). A pathway analysis was performed to elucidate the significant pathways of DEGs using the Kyoto Encyclopedia of Gene and Genomes (KEGG) (http://www.genome.jp/kegg, accessed on 20 January 2022).

### 4.5. Metabolite Identification and Quantification

The Wuhan MetWare Biotechnology Co., Ltd. (Wuhan, China) identified and quantified the metabolites. The metabolite data acquisition instrument system mainly includes ultraperformance liquid chromatography, UPLC (SHIMADZU Nexera X2^1^), and tandem mass spectrometry, MS/MS (Applied Biosystems 4500 QTRAP^2^). Liquid phase conditions were as follows: HPLC column, Agilent SB-C18 1.8 µm, 2.1 mm × 100 mm; solvent system, phase A is ultrapure water (adding 0.1% formic acid), phase B is acetonitrile (adding 0.1% formic acid); gradient program, phase B is 95:5 *v/v* at 0.00 min, 5:95 *v/v* at 10.0 min, 5:95 *v/v* at 11.0 min, 95:5 *v/v* at 12.1 min, 95:5 *v*/*v* at 14.0 min; flow rate, 0.35 mL/min; temperature, 40 °C; injection volume: 4 µL. The mass spectrum conditions mainly include: LIT and triple quadpole (QQQ) scans were obtained on a triple quadpole linear ion trap mass spectrometer (Q Trap), and AB4500 Q Trap UPLC/MS/MS system equipped with an ESI turbo ionspray interface. It can be controlled by analyst 1.6.3 software (AB Sciex) to run both positive and negative ion modes. The ESI source operation parameters are as the follows: ion source turbo spray; source temperature 550 °C; ion spray voltage (IS) 5500 V (Positive ion mode)/−4500 V (negative ion mode); ion source gas I (GSI), gas II (GSII), and curtain gas (CUR) set at 50, 60, and 25.0 psi, respectively; and high collision gas (CAD). The instrument was tuned and calibrated with a 10 and 100 µmol/L polypropylene glycol solution in QQQ and LIT modes, respectively. The QQQ scans were acquired as multiple reaction monitoring (MRM) experiments with the collision gas (nitrogen) set to medium. Through further optimization of DP and CE, the DP and CE of each MRM ion pair were completed. A specific set of MRM ion pairs was monitored in each period based on the metabolites eluted in each period. The filtering conditions for differentially accumulated metabolites (DAMs) were as follows: absolute log 2 (fold change) ≥ 1, *p*-value < 0.05, and variable importance in projection (VIP) ≥ 1. To study the accumulation of specific metabolites, a principal component analysis (PCA) and orthogonal partial least squares discriminant analysis (OPLS-DA) were performed using R (www.r-project.org/, accessed on 23 January 2022).

### 4.6. GO and KEGG Pathway Enrichment Analysis

A GO function analysis of differentially expressed genes was performed by GOseq, including GO function enrichment and GO function clustering of differentially expressed genes. The database used was the Gene Ontology database (http://www.geneontology.org/, accessed on 25 January 2022). The KEGG enrichment analysis of differentially expressed genes and differential metabolites was performed using the KOBAS software and KEGG (http://www.genome.ip/kegg/, accessed on 25 January 2022) database.

### 4.7. Real-Time Fluorescence Quantitative Analysis

To verify the accuracy of the DEG data obtained by the RNA seq, the relative expression levels of four key DEGs were analyzed by qRT-PCR. Fluorescent quantification primers for DEGs and *Brassica napus Actin2.1* (internal control) were designed using Primer 5.0 and they are listed in Appendix A. The relative expression was calculated according to the method of Livak and Schmittgen [49]. The RNA was extracted from the samples (G218 and G230) and was used to synthesize first-strand cDNA by using TransScript First-Strand cDNA Synthesis SuperMix (TransGene, AT301-02) following the manufacturer’s instructions. qRT-PCR was performed using PerfectStart^®^ Green qPCR SuperMix (TransGene, AQ601-02) according to the manufacturer’s instructions. The PCR conditions were as follows: 95 °C for 30 s denaturation, 40 cycles of 94 °C denaturation for 5 s, and 60 °C annealing and extension for 30 s. In this study, all the genes were repeated in three biological replicates (each biological replicate contains three technical replicates). The 2^−∆∆Ct^ method was used to calculate the mRNA expression level of genes. The relative gene expression level and FPKM were normalized by using log2 (fold change) measurements.

### 4.8. External Application of Vitamin B6

To verify the effect of vitamin B6 metabolic pathways on rapeseed under continuous waterlogging stress, one hour before the potting simulated waterlogging stress (the method is the same as Section 4.1), the exogenous vitamin B6 was evenly sprayed on the leaves of seedlings at the five-leaf stage with a spray pot, and the phenotype was observed. The experiment was repeated three times, the spray concentration was 1 mg/L, and the vitamin B6 reagent was purchased from Coolaber (Beijing, China).

## 5. Conclusions

This study screened a waterlogging-tolerant *Brassica napus* inbred line G230 and waterlogging-intolerant *Brassica napus* inbred line G218 by simulating waterlogging stress and field waterlogging stress. The effects of waterlogging stress on the two inbred lines were analyzed by transcriptomics and metabolomics. A comprehensive analysis of rapeseed’s root and leaf transcriptomes and metabolomes indicated that flavonoids and pyridoxal phosphate might play important roles in the waterlogging tolerance of *Brassica napus*. Additionally, some genes, such as *CHI*, *DRF*, *LDOX*, *PDX1.1*, and *PDX2* may be the key genes for waterlogging tolerance in *Brassica napus.* Moreover, found that external application of vitamin B6 can effectively improve the waterlogging tolerance of rape.

## Figures and Tables

**Figure 1 ijms-24-06015-f001:**
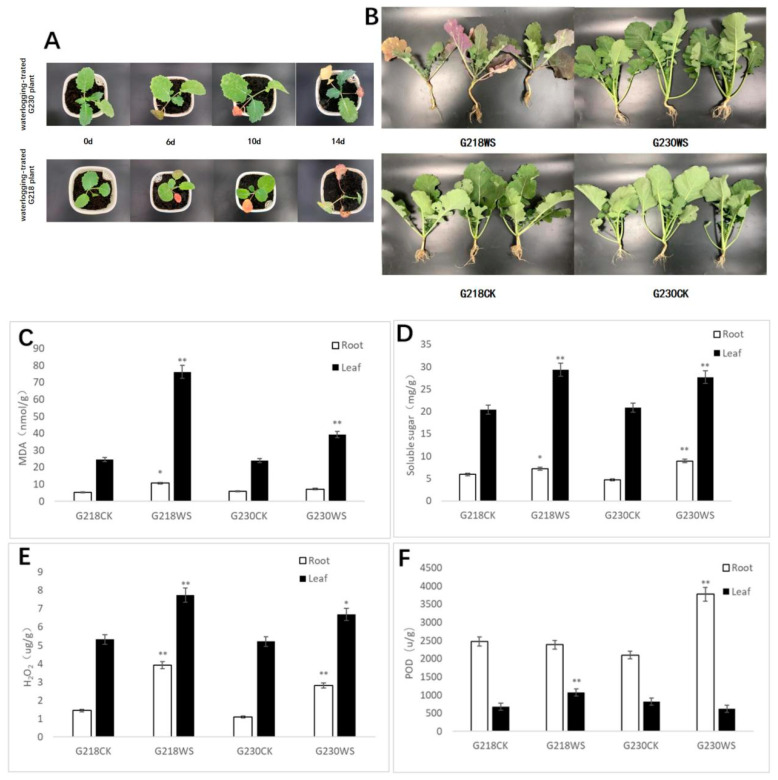
Morphological and physiological changes of rapeseed after waterlogging stress. (**A**) Potted simulated field waterlogging experiment. (**B**) Rape plants reoxygenated for seven days after six days of waterlogging stress in the field. (**C**) Changes in MDA content. (**D**) Changes in soluble sugar content. (**E**) Changes in H_2_O_2_ content. (**F**) Changes in POD content. * indicates a significant difference (*p* < 0.05); ** indicates a very significant difference (*p* < 0.01). G218CK is the control treatment of G218, and G218WS is the waterlogging stress treatment of G218; G230CK is the control treatment of G230, and G230WS is the waterlogging stress treatment of G230.

**Figure 2 ijms-24-06015-f002:**
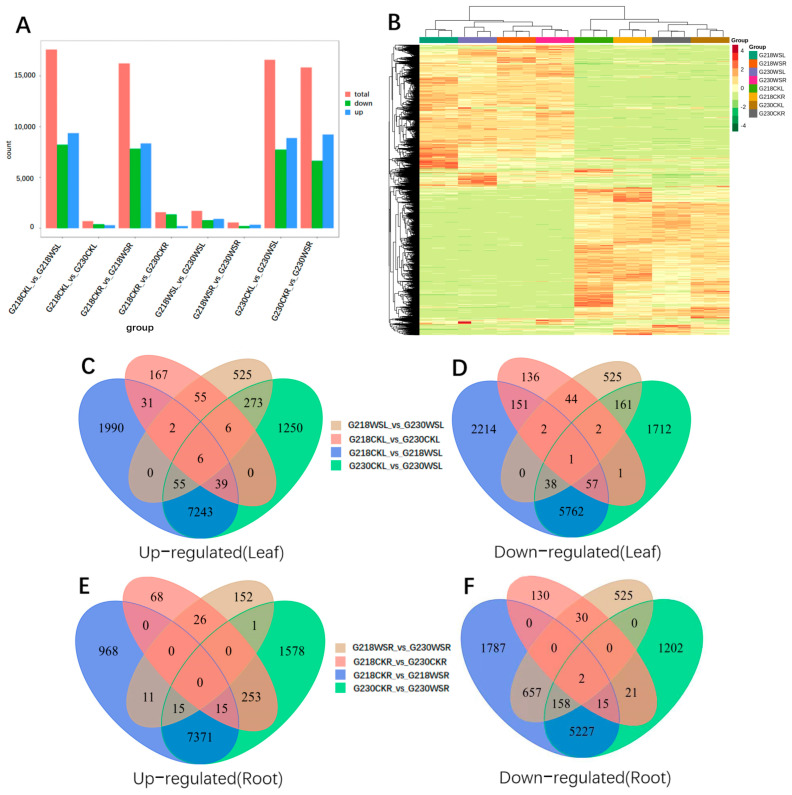
Analysis of DEGs under waterlogging stress. (**A**) Up- and down-regulated DEGs in different comparisons. (**B**) Heatmap of DEGs. (**C**) Venn diagram of up-regulated DEGs in comparative combinations of leaf sample sections. (**D**) Venn diagrams of down-regulated DEGs in a comparative combination of leaf sample sections. (**E**) Venn diagram of up-regulated DEGs in fractions of root samples comparing combinations. (**F**) Venn diagrams of down-regulated DEGs in partial comparison combinations of root samples.

**Figure 3 ijms-24-06015-f003:**
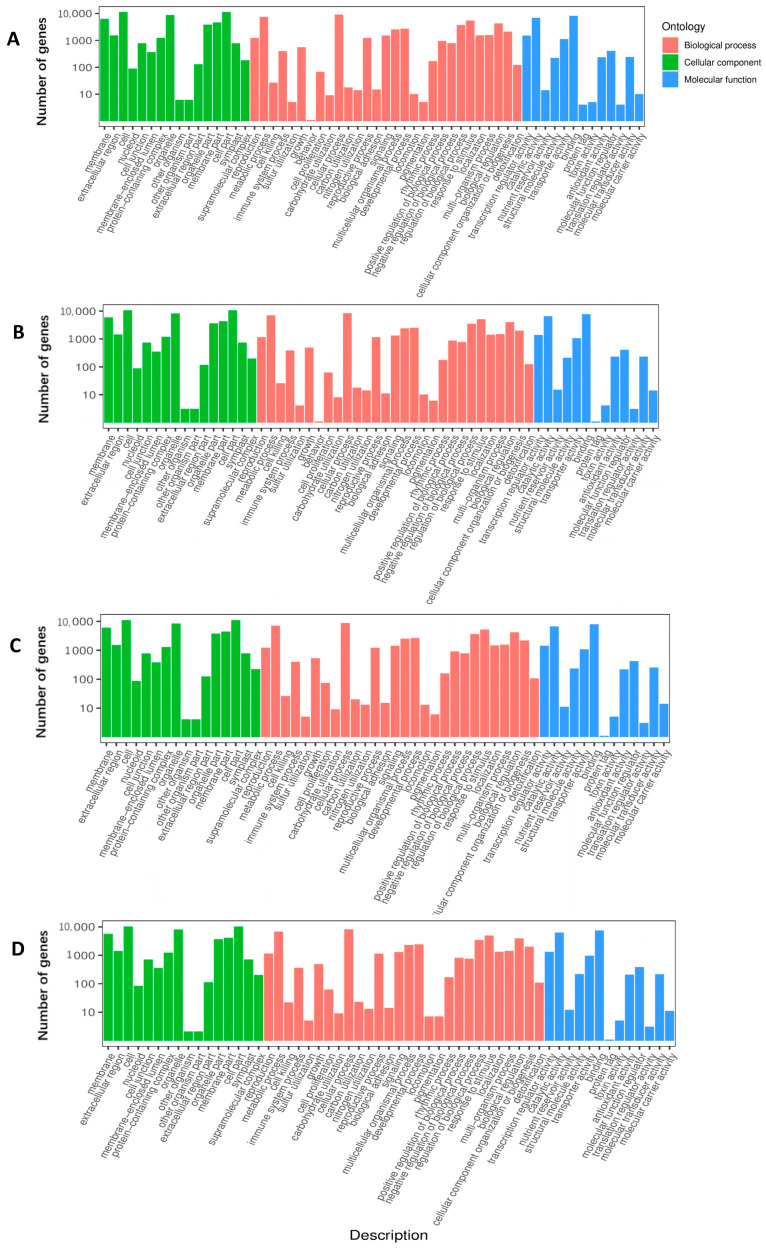
GO enrichment analysis of DEGs in different comparative combinations under waterlogging stress. (**A**) Top 50 GO Terms with the most enriched DEGs in the G218CKL vs. G218WSL comparison. (**B**) Top 50 GO Terms with the most enriched DEGs in the G218CKR vs. G218WSR comparison. (**C**) Top 50 GO Terms with the most enriched DEGs in the G230CKL vs. G230WSL comparison. (**D**) Top 50 GO Terms with the most enriched DEGs in the G230CKR vs. G230WSR comparison.

**Figure 4 ijms-24-06015-f004:**
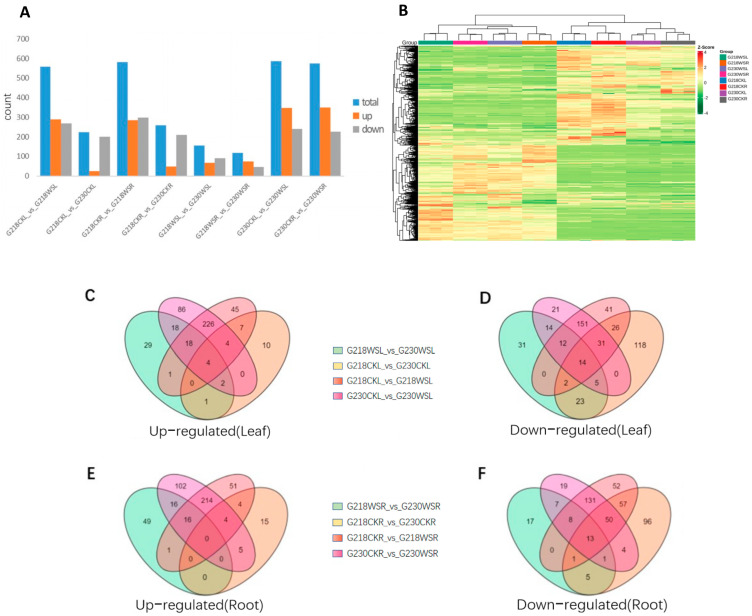
Analysis of DEMs under waterlogging stress. (**A**) Up- and down-regulated DEMs in different comparisons. (**B**) Heatmap of DEMs. (**C**) Venn diagram of up-regulated DEMs in comparative combinations of leaf sample sections. (**D**) Venn diagrams of down- and up-regulated DEMs in a comparative combination of leaf sample sections. (**E**) Venn diagram of up-regulated DEMs in fractions of root samples comparing combinations. (**F**) Venn diagrams of down-regulated DEMs in partial comparison combinations of root samples.

**Figure 5 ijms-24-06015-f005:**
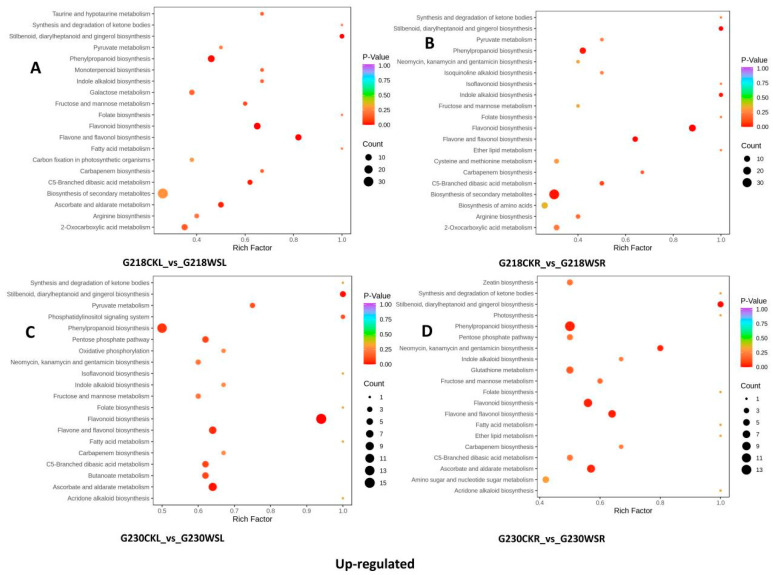
KEGG enrichment analysis of upregulated DEMs of different comparative combinations under waterlogging stress. (**A**) Significantly upregulated KEGG pathways in the G218CKL vs. G218WSL comparison. (**B**) Significantly upregulated KEGG pathways in the G218CKR vs. G218WSR comparison. (**C**) Significantly upregulated KEGG pathways in the G230CKL vs. G230WSL comparison. (**D**) Significantly upregulated KEGG pathways in the G230CKR vs. G230WSR comparison.

**Figure 6 ijms-24-06015-f006:**
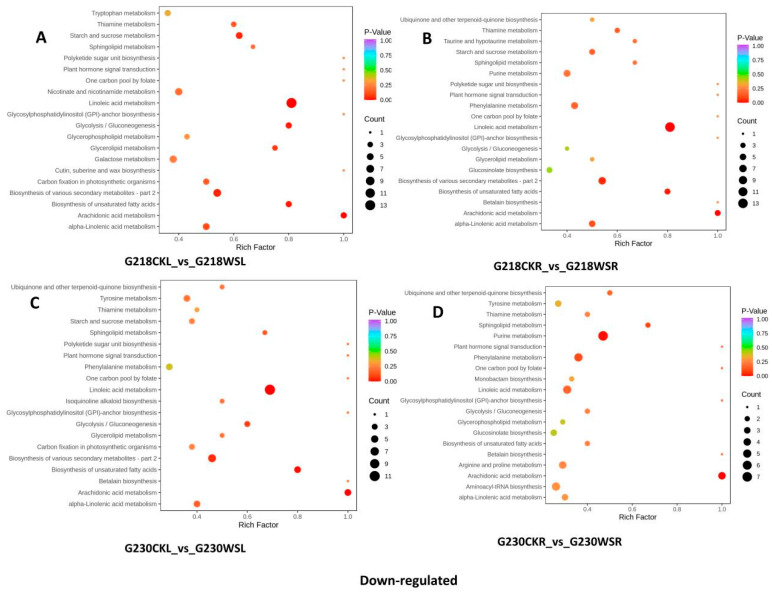
KEGG enrichment analysis of downregulated DEMs of different comparative combinations under waterlogging stress. (**A**) Significantly downregulated KEGG pathways in the G218CKL vs. G218WSL comparison. (**B**) Significantly downregulated KEGG pathways in the G218CKR vs. G218WSR comparison. (**C**) Significantly downregulated KEGG pathways in the G230CKL vs. G230WSL comparison. (**D**) Significantly downregulated KEGG pathways in the G230CKR vs. G230WSR comparison.

**Figure 7 ijms-24-06015-f007:**
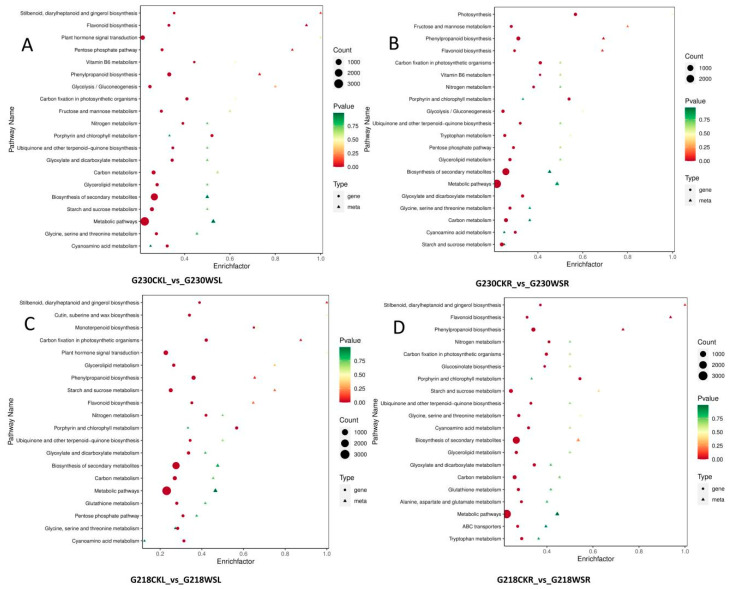
Combined analysis of DEGs and DEMs under waterlogging stress. (**A**) Analysis of DEGs and DEMs enriched in the same pathway in G230CKL vs. G230WSL comparison. (**B**) Analysis of DEGs and DEMs enriched in the same pathway in G230CKR vs. G230WSR comparison. (**C**) Analysis of DEGs and DEMs enriched in the same pathway in G218CKL vs. G218WSL comparison. (**D**) Analysis of DEGs and DEMs enriched in the same pathway in G218CKR vs. G230WSR comparison.

**Figure 8 ijms-24-06015-f008:**
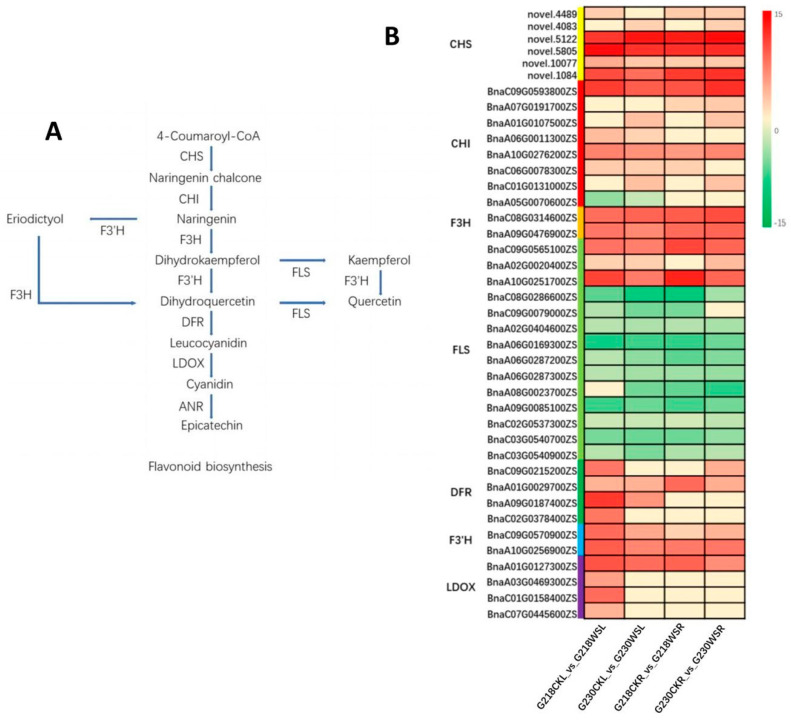
Transcriptome analysis of flavonoid biosynthetic pathways. (**A**) Flavonoid biosynthesis pathway. (**B**) Differential expression of key genes involved in flavonoid biosynthesis pathway; the heatmap scale ranges from −15 to +15 on a log_2_ fold change.

**Figure 9 ijms-24-06015-f009:**
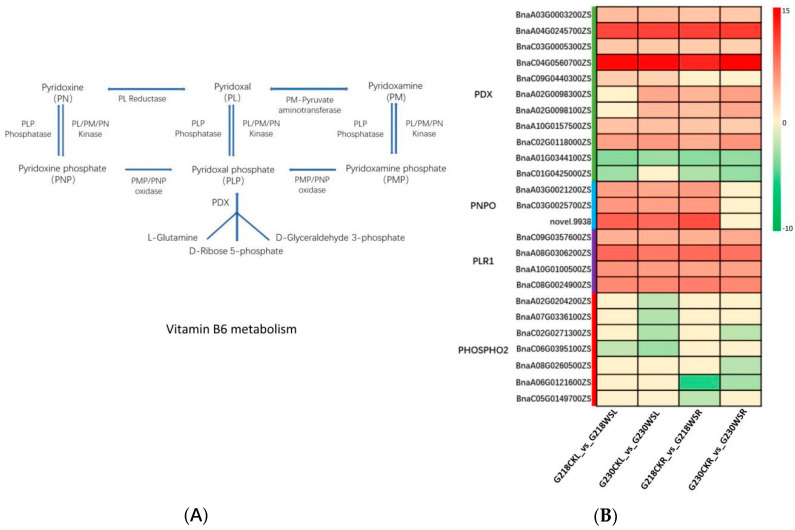
Transcriptome analysis of vitamin B6 metabolism. (**A**) Vitamin B6 metabolic pathway. (**B**) Differential expression of key genes involved in vitamin B6 metabolic pathways; the heatmap scale ranges from −10 to +15 on a log_2_ fold change.

**Figure 10 ijms-24-06015-f010:**
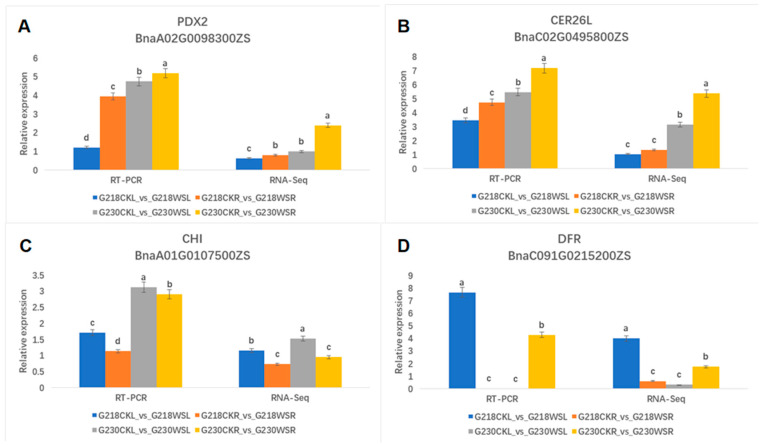
Comparison of expression levels of key genes in RNA-seq and qRT-PCR analysis. (**A**) *PDX2*, (**B**) *CER26L*, (**C**) *CHI*, and (**D**) *DFR* of *Brassica napus* of G218 and G230 under waterlogging stress conditions. Data are the means of three replicates ± SD, and the different letters (a–d) indicate a significant difference at *p* ≤ 0.05 according to Duncan’s test.

**Figure 11 ijms-24-06015-f011:**
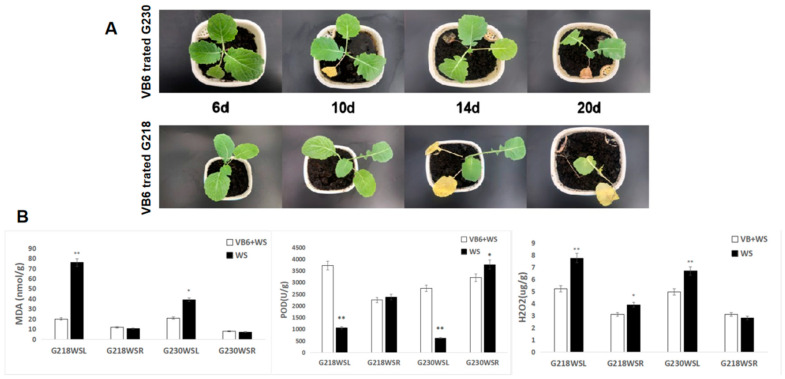
Effect of exogenous VB6 on G218 and G230. (**A**) Effects of foliar sprays of vitamin B6 on the phenotype of rapeseed under simulated waterlogging stress. (**B**) Changes of MDA, POD, and H_2_O_2_ in leaves and roots of G218 and G230 under waterlogging stress after VB6 application. * indicates a significant difference (*p* < 0.05); ** indicates a very significant difference (*p* < 0.01).

**Table 1 ijms-24-06015-t001:** Agronomic traits analysis of rape under waterlogging stress.

Trait	G218		G230		WTC	
	CK	WS	CK	WS	G218	G230
Root length (cm)	356.56 ± 41.49	190.32 ± 54.45 **	211.08 ± 14.44	260.72 ± 16.03	0.53	1.24 **
Tips	1068.33 ± 63.99	743 ± 66.88 **	828.67 ± 21.55	970.67 ± 82.93	0.69	1.17 **
The shoot fresh weight (g)	54.73 ± 5.09	24.58 ± 4.06 **	54.62 ± 6.65	48.82 ± 3.20	0.45	0.89 **
The root fresh weight (g)	3.47 ± 0.75	2.56 ± 0.55 **	3.52 ± 0.57	6.45 ± 1.20 **	0.74	1.83 **
The total fresh weight (g)	58.20 ± 5.84	27.14 ± 4.61 **	58.14 ± 7.22	55.27 ± 4.40	0.47	0.95 **

In the table, CK is the control treatment; WS is the waterlogging stress treatment; ** indicates a very significant difference (*p* < 0.01). WTC is waterlogging tolerance coefficient.

**Table 2 ijms-24-06015-t002:** Several metabolites induced by waterlogging stress in flavonoid biosynthesis and vitamin B6 metabolic pathways.

Metabolic Pathways	Metabolites	FC			
		G218CKL_vs_G218WSL	G230CKL_vs_G230WSL	G218CKR_vs_G218WSR	G230CKR_vs_G230WSR
Flavonoid biosynthesis	Naringenin chalcone	1.188	1.619	1.506	NA
	Naringenin	NA	1.445	1.892	NA
	Epiafzelechin	NA	11.546	11.427	NA
	Eriodictyol	10.766	10.253	10.113	10.378
	Dihydroquercetin	4.563	NA	6.031	6.384
Vitamin B6 metabolism	Pyridoxal	2.576	2.060	NA	1.100
	Pyridoxal phosphate	−14.201	NA	−13.812	NA
	Pyridoxine	−2.486	−1.167	−3.118	−2.034

FC: fold change. NA: |fold change| < 1.

**Table 3 ijms-24-06015-t003:** Grouping of experimental materials.

Treatment	G218	G230
Control check (leaves)	G218CKL	G230CKL
Waterlogging stress (leaves)	G218WSL	G230WSL
Control check (roots)	G218CKR	G230CKR
Waterlogging stress (roots)	G218WSR	G230WSR

## Data Availability

The data presented in this study are available on request from the corresponding author.

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
