# Peer review of "Tissue-Specific Transcriptome and Metabolome Analysis Reveals the Response Mechanism of Brassica napus to Waterlogging Stress"

_ijms, 2023, doi:10.3390/ijms24076015_

Round 1
Reviewer 1 Report
The review is enclosed as attachment.

Author Response
Thank you for your suggestions and comments, which will help us to make our manuscripts better. The following content is our response to your suggestions and comments.
Point 1: Why were samples for laboratory tests taken after 7 days of recovery? In most publications on waterlogging stress tolerance mechanisms, samples are taken at various stages just during stress. Of course, it is important to study the recovery period after stress, but the mechanisms are related to the process of recovery after stress (especially since it was a long recovery period - 7 days), and not to stress tolerance. In the publication, the authors did not justify the choice of this date of sampling for analyses.
Response 1: There are two reasons why samples under waterlogging stress for 6 days and reoxygenated for 7 days were selected for transcriptome and metabolome analysis. On the one hand, the phenotypic difference between G230 and G218 was obvious at this time point. The leaves of G218 turned yellow or purple, while the leaves of G230 did not change significantly. On the other hand, physiological data showed that there were significant differences between G230 and G218 in reactive oxygen scavenging ability. Therefore, sampling at this point in time for transcriptome and metabolome analysis can explore which genes and metabolites are involved.
Point 2: The course of the experiment is not sufficiently described: no description of the experimental setup / how many repetitions / blocks, cultivation conditions (substrate, light, ....), how the stress was induced, how the state of hypoxia was monitored (especially that in field conditions it was not it is obvious).
Response 2: We agree with you. In the resubmitted manuscript, we have made supplements in the section of materials and methods and marked them in red (paragraph 4.2).
Point 3: Were samples taken from greenhouse or field cultivation? It’s not clear.
Response 3: I am very sorry that we did not explain this point clearly in the manuscript. The sample was taken from the field. We have supplemented this in the resubmitted manuscript (paragraph 4.2).
Point 4: Incorrect description in table 3 - treatment: control (leaves)?
Response 4: This is our writing error. What we want to express is the leaves of the control check. We revised this in the resubmitted manuscript.
Point 5: Description of the methodology in section 4.2 - insufficient, no name of kit? Was it
single kit for MDA analysis, POD hydrogen peroxide? I don't think there is such a kit.
Response 5: Each indicator has a unique content kit. In the resubmitted manuscript, we revised it (paragraph 4.3).
Point 6: No description of the methodology for the results presented in Table 1.
Response 6: Root length and tips were analyzed by root scanner. The waterlogging tolerance coefficient is the ratio of the measured value of the treatment to the measured value of the control check. We have supplemented in the resubmitted manuscript ( paragraph 4.3).
Point 7: In paragraph 4.3 - no information regarding sequencing parameters i.e. Paired ends or
single end mode?
Response 7: In paragraph 4.4 of the revised manuscript, we have described the sequencing parameters in detail.
Point 8: Section 4.4. – please provide details of the parameters for the identification of
metabolites.
Response 8: In paragraph 4.5 of the revised manuscript, we have described the sequencing parameters in detail.
Point 9: Paragraph 4.6 - no description of detailed qPCR conditions (composition of the reaction mixture and temperature profile, no temperature of denaturation for primers and length of products - table S1). Primers are not designed on NCBI - this is not a primer design program .
Response 9: In paragraph 4.7 of the revised manuscript, we have described the details of qPCR. temperature of denaturation for primers and length of products have been added to Table S4. As for the design of primers, we found the gene sequence on NCBI, and then designed the primers through Primer 5.0.
Point 10: Section 4.7. – no details on the application of vitamin B6 i.e. at what time, how the spraying was carried out.
Response 10: One hour before the simulated waterlogging stress of potted plants, the exogenous vitamin B6 was evenly sprayed on the leaves of seedlings at the five-leaf stage with a spray pot. We have described this in the resubmitted manuscript ( paragraph 4.8).
Point 11: All figures are completely illegible, in particular Figs. 2-9. Fig. S3, Fig. S4 are not included in supplementary materials, there are errors in the description of figures i.e.Fig. 4B-F there is erro– instead of DEMs there is DEGs, Fig. 2D–error - should be only down-regulated and there is up- and down-regulated DEGs.
Response 11: In the resubmitted manuscript, we have tried to improve the clarity of the figures, and figure S4 have been added to the supplementary material. All incorrect legends have been changed.
Point 12: The description of the results is too extensive, sometimes it is difficult to find out what
the results refer to. Maybe it's worth combining the results of DEGs with DEMs?
Response 12: I'm sorry, we can't find a better way to combine DEGs and DEMs, and we have re-described the results in the conclusion, hoping that readers can see the results clearly.
Point 13: In Abstract: line 23 - there is: …were more up-regulated… - such a term cannot be used,
only the level of gene expression can be higher or lower.
Response 13: We revised this in the resubmitted manuscript.
Point 14:
- Lines 76-77 - based on morphology, writing that a given variety is tolerant and the other sensitive is too bold.
Response: We reached this conclusion through morphological and physiological data. This has been revised in the resubmitted manuscript.
- Line 79 - …the field water content was greater than the maximum soil water content … - please add these results as supplementary material.
Response: The results have been placed in Figure S1 of supplementary materials.
- Table 1 - in column 1 use full parameter names instead of abbreviations, varieties
should be side by side in the control and stress - for easier comparison.
Response: In the resubmitted manuscript, we have adjusted Table 1.
- Table S2 - results for Clean bases are not presented correctly - there are dates.
Response: A in table S2 is a shorthand that has a count mark (G) that resembles a unit.
- Line 146 - the number of DEGs alone does not prove tolerance to stress - Conclusion formulated exaggerated.
Response: After careful consideration, we think you are right, so we have removed this sentence.
- No description of gene expression results - it is not known what genes they are, writing: "The expression patterns of DEGs obtained by RNA-seq and qRT-PCR were highly consistent, indicating the reliability of RNA-seq results(Fig. 10A).” - on what basis is this statement? No correlation results, no statistical analysis results.
Response: We agree with you and have revised it in the resubmitted manuscript.(Fig.10, paragraph 2.9 )
- When validating the effect of external treatment of vitamin B6 to improve waterlooging stress, it is not enough to rely on morphology alone, but it is also possible to analyze the expression of genes by qPCR related to the metabolism of vitamin B6, or at least analyze physiological parameters.
Response: We agree with you and have added some physiological parameters in the resubmitted manuscript.(Fig.11, paragraph 2.10 ).

Reviewer 2 Report
Manuscript No. IJMS-2211445
Title: Tissue-specific transcriptome and metabolome analysis reveals the response mechanism of Brassica napus to waterlogging stress
Comments:
# The authors of this research have tried to screen a waterlogging-tolerant Brassica napus inbred line G230 and waterlogging-intolerant Brassica napus inbred line G218 initially by simulating waterlogging stress and field waterlogging stress. The stress responses of these two contrasting inbred lines were analyzed through morphological and physiological or biochemical parameters. Further, they compared these lines based on transcriptomics and metabolomics of root and leaf. Based on comprehensive analysis, they opined that flavonoids and pyridoxal phosphate might play important roles in the waterlogging tolerance of Brassica napus. They also found that external application of vitamin B6 could effectively improve the waterlogging tolerance of rape. However, I would forward some queries that are needed to be clarified before making any conclusion on the vital role of vit. B6 metabolic pathways in waterlogging stress tolerance.
# MDA content under waterlogging stress has increased more significantly in susceptible line G218 as compared to tolerant line G230. Therefore, it would be assumed that more oxidative damage has occurred in G218 than G230. It is also possible that G230 is capable to produce more antioxidants to minimize the oxidative damages. Thus, it would have been better if the comprehensive analysis of overall antioxidative systems is done rather than only the two antioxidant-related pathways of flavonoid biosynthesis and vitamin B6 metabolism. And then only, certain conclusion can be made on the role of only these two antioxidative systems in waterlogging stress tolerance of Brassica napus.
# Pg 2/line 53-55----- “to screen more waterlogging-tolerant rapeseed varieties and elucidate their potential resistance mechanisms, some rapeseed varieties and inbred lines were selected for pot and field cultivation, and the phenotypes of tolerant and susceptible rapeseed”—how many varieties were screened out? Though, as per manuscript, the authors had worked with two inbred lines only.
# Pg 2/line 78-79----“in field cultivation, we subjected these two materials to field waterlogging stress (the field water content was greater than the maximum soil water content) for six days at the seedling stage (5 leaves) and then reoxygenated for seven days”--- how waterlogging stress water was created? Or how soils were reoxygenated? Please clarify these in materials and methods section since it is not very clear.
# The manuscript is found to be unnecessarily long and it should be reduced since there are many repetitions.
# The images and graphs (particularly legends) given in the manuscript are very hazy and not clear, and these need to be improved.
# I would also suggest a revision of the English language.
Author Response
Thank you for your suggestions and comments, which will help us to make our manuscripts better. The following content is our response to your suggestions and comments.
Point 1: MDA content under waterlogging stress has increased more significantly in susceptible line G218 as compared to tolerant line G230. Therefore, it would be assumed that more oxidative damage has occurred in G218 than G230. It is also possible that G230 is capable to produce more antioxidants to minimize the oxidative damages. Thus, it would have been better if the comprehensive analysis of overall antioxidative systems is done rather than only the two antioxidant-related pathways of flavonoid biosynthesis and vitamin B6 metabolism. And then only, certain conclusion can be made on the role of only these two antioxidative systems in waterlogging stress tolerance of Brassica napus.
Response 1: Antioxidant system, including flavonoid biosynthesis, vitamin B6 metabolism, glutathione metabolism, ascorbic acid metabolism and other pathways. In fact, we found that after waterlogging stress, the genes and metabolites related to flavonoid biosynthesis and vitamin B6 metabolism pathway in G230 were significantly enriched, rather than other antioxidant pathways, through the combined analysis of transcriptome and metabolome.
Point 2: Pg 2/line 53-55----- “to screen more waterlogging-tolerant rapeseed varieties and elucidate their potential resistance mechanisms, some rapeseed varieties and inbred lines were selected for pot and field cultivation, and the phenotypes of tolerant and susceptible rapeseed”—how many varieties were screened out? Though, as per manuscript, the authors had worked with two inbred lines only.
Response 2: One study found a rapid screening method for waterlogging tolerant rapeseed. We screened 32 rapeseed varieties or strains by this method, and found that G230 was waterlogging tolerant and G218 was waterlogging sensitive. We have supplemented this part in the resubmission of the manuscript.(Paragraphs 2.1 and 4.1)
Point 3: Pg 2/line 78-79----“in field cultivation, we subjected these two materials to field waterlogging stress (the field water content was greater than the maximum soil water content) for six days at the seedling stage (5 leaves) and then reoxygenated for seven days”--- how waterlogging stress water was created? Or how soils were reoxygenated? Please clarify these in materials and methods section since it is not very clear.
Response 3: The water under waterlogging stress in the field is artificially irrigated into the test field where the water outlet is blocked. After the end of flooding, the water outlet is opened to release oxygen. In the resubmitted manuscript, we made additional explanations.(Paragraphs 4.2)
Point 4: The manuscript is found to be unnecessarily long and it should be reduced since there are many repetitions.
Response 4: In the resubmitted manuscript, we have deleted some duplicate contents.
Point 5: The images and graphs (particularly legends) given in the manuscript are very hazy and not clear, and these need to be improved.
Response 5: In the resubmitted manuscript, we have improved the clarity of the pictures.
Point 6: I would also suggest a revision of the English language.
Response 6: We asked native English speakers to edit the resubmitted manuscript for language.

Round 2
Reviewer 1 Report
Please follow the file that I attached. In red colour I wrote my answer to your reply

Author Response
Thank you for your comments. Please follow the file that I attached. In deep red colour I wrote my answer to your reply.
